# Acute Myeloid Leukemia Evolving from Myeloproliferative Neoplasms: Many Sides of a Challenging Disease

**DOI:** 10.3390/jcm10030436

**Published:** 2021-01-23

**Authors:** Francesco Mannelli

**Affiliations:** 1SOD Ematologia, Università di Firenze, AOU Careggi, 50134 Firenze, Italy; francesco.mannelli@unifi.it; 2Centro Ricerca e Innovazione Malattie Mieloproliferative (CRIMM), AOU Careggi, 50134 Firenze, Italy

**Keywords:** acute myeloid leukemia, myeloproliferative neoplasms, blast phase, secondary acute lekemia

## Abstract

The evolution to blast phase is a frequently unpredictable and almost invariably fatal event in the course of myeloproliferative neoplasms. The molecular mechanisms underlying blast transformation have not been elucidated and the specific genetic and epigenetic events governing leukemogenesis remain unclear. The result of the long-lasting dynamics, passing through progressive genetic steps, is the emergence of one or more clones often characterized by complex genetics, either at conventional karyotyping or at modern high-throughput sequencing analyses, with all clinical and prognostic correlates. The current therapeutic approaches are largely inadequate and incapable of modifying the inherent unfavorable outcome. In this perspective, the application of targeted strategies should aim to prevent the occurrence of leukemic evolution. At transformation, the crucial target of treatment should be the allocation to allogeneic transplant for eligible patients. With this in mind, novel combination treatments may provide useful bridging strategies, beyond potentially improving outcomes for patients who are not candidates for intensive approaches.

## 1. Introduction

Myeloproliferative neoplasms (MPN) are clonal disorders of hematopoietic stem cells that include polycythemia vera (PV), essential thrombocytemia (ET), and primary myelofibrosis (PMF). A subset of MPN patients transform to secondary acute myeloid leukemia (AML), generally defined as blast phase (MPN-BP). Although some treatments used in the past, such as chlorambucil, 32P, and pipobroman, were shown to increase the rate of leukemic transformation, no compelling evidence supporting an association of therapy with AML has been produced for hydroxyurea, the mostly used drug, interferon, and anagrelide or JAK2 inhibitors more recently. AML occurs approximately in 5–10% of cases, being more frequent in PMF (20% of patients over a 10-year period) compared to PV (4%) and ET (1%) [1], the latter evolution rate likely to be reduced further when considering “true” ET as separated from pre-fibrotic PMF [2]. According to the World Health Organization (WHO), MPN-BP is defined by the presence of a minimum of 20% blasts in either peripheral blood (PB) or bone marrow (BM) at morphological analysis in a patient with an underlying MPN [3]. Most patients evolving from chronic-phase MPN to AML experience a transition through an accelerated phase (AP), featured by a lower percentage of blasts (10% to 19% on PB or BM), although leukemic transformation can also occur more abruptly with a quick raise in blast percentage [4].

## 2. Biological Features

The molecular mechanisms underlying leukemic transformation have not been elucidated yet. That is likely due to the large genetic heterogeneity of MPN with several concomitant mutations, possibly involving co-existing clones. A further effect depends on the order on which genetic abnormalities are acquired, each one shaping a specific genetic and epigenetic background that influences the probability and the consequences of the subsequent one(s). Therefore, the definition of a model of disease transformation to MPN-BP has turned out to be extremely challenging.

In addition to driver mutations (involving *JAK2*, *CARL* and *MPL* genes), which affect the clinical phenotype of MPN, several further mutations have been described throughout the course of the disease. In order to shed light on the evolution process, Grinfeld et al. proposed a molecular classifier that simultaneously takes into account the multiplicity and interaction of molecular abnormalities, in order to provide a personalized prediction of the probability of transformation [5].

In general terms, the molecular pathway leading to MPN-BP can proceed through at least two distinct routes. First, *JAK2/CALR/MPL*-positive MPN progresses to *JAK2/CALR/MPL*-positive AML, in turn associated with the acquisition of additional genetic alterations. On the other side, an MPN can go through loss of pre-existing driver mutation at AML onset, probably due to the transformation of an antecedent or de novo mutant clone leading to MPN-BP that overcomes the MPN *JAK2/CALR/MPL*-mutated clone [6,7,8].

Among additional mutations, those affecting *TP53* often coincide with MPN-BP, are synergistic with *JAK2* mutation on leukemogenesis, and have been associated with a slower, long-term transformation [9]. Mutations involving *TET2* typically predate MPN drivers [10] but they can occur in subsequent disease phases as well, including at transformation [11]. Mutations occurring in *RUNX1*, *IDH1/2*, and *U2AF1* have been preferentially described in cases with more rapid leukemic progression [9]. In contrast, *ASXL1* mutations have been reported at all phases of disease, suggesting a specific contribution in clonal evolution [11]. Abnormalities involving long arm of chromosome 1 have been associated with clonal progression and evolution to MPN-BP, possibly due to overexpression of MDM4, a negative regulator of TP53 [12], further underlying the role of the latter in this context. Mutations in RNA splicing factors (*SF3B1*, *U2AF1*, *SRSF2*, and *ZRSR2*) are relatively frequent in MPN-BP, with a similar incidence to that observed in the chronic phase of MPN. Their occurrence is generally associated with later phases of MPN course [13].

The application of next generation sequencing (NGS) technique is becoming increasingly important in MPN, especially MF, and in patients potentially eligible for allogeneic transplant, as recommended by modern genetic-based prognostic models [14]. Although not routinely applied, NGS can add useful information at the onset of MPN-BP, as specific genotypes may benefit from different and targeted approaches.

The complex genetic background often translates into peculiar morphological and phenotypic profiles. Blasts from MPN-BP most often show myeloid phenotype with erythroid or megakaryocytic lineage differentiation, relatively uncommon in de novo AML, as suggested by a higher frequency of M6 and M7 subtypes according to the historical FAB classification [15,16].

## 3. Clinical Peculiarities and Prognosis

MPN-BP is associated with dismal outcome, with median survival of 3–6 months from diagnosis. Currently, there are limited published prospective therapeutic trials in patients with MPN-BP, and only a few retrospective reviews of institutional experience. There is an unmet need for clinical trials with innovative approaches for the treatment of patients with MPN-BP. Main statements about therapeutic strategies are derived and adapted from other clinical contexts, i.e., adverse karyotype or primary refractory de novo AML or secondary to MDS. The extremely unfavorable prognosis is consistent with some specific features.

First, advanced age at diagnosis and relative comorbidities often limits the access to aggressive treatment approaches [17]. Biologically, the long-lasting dynamics of clone selection and evolution preceding AML makes it one of the most challenging diseases in onco-hematology. Leukemic cells usually display unfavorable genetic abnormalities (i.e., high-risk karyotype and/or *TP53* mutations) with resulting low probability of response to conventional chemotherapy. Beyond frequent primary resistance of blasts, the co-existence of underlying MPN further complicates the clinical management and makes response to treatment according to standard criteria (ELN 2017) quite rare and short-lived when occurring. Complete remission (CR) to standard (i.e., “3 + 7”) or intensified (i.e., FLA-Ida, MEC) induction schemes is reported to range from 0% to 30%, often with incomplete hematopoietic recovery (CRi) [16,18,19]. In fact, the co-existence of MPN and its unique genotypic profile have led researchers to delineate specific response criteria suited to assess therapeutic response of MPN-BP [20]. These criteria mandate for assessment of both components of the disease, which are MPN and BP, and require the evaluation of clinical (spleen), morphological (PB and BM smears), and molecular markers associated with either MPN or BP (Table 1).

Available evidence from the literature shows that allogeneic hematopoietic stem cell transplant (HSCT) is the only recognized therapeutic option potentially able to achieve the response categories of complete molecular response (CMR) and complete cytogenetic response (CCR) by consortium criteria.

Induction death rate is high due to prolonged aplasia, the risk of infection further enhanced by MPN-related immunosuppression, and low return from transfusion support caused by organomegaly (especially with preceding PMF).

According to published evidence, adverse karyotype and molecular genetics show some pejorative effect on outcome of MPN-BP. When available, karyotype is abnormal in the majority of cases, ranging from 64% to 92.7% [16,19,21,22,23,24]. On the basis of these abnormalities, 30–60% of cases are classified as adverse risk according to main stratification systems [25,26]. Exceptional cases with CBF rearrangements, translocation t(15;17)/*PML*-*RARα* or *NPM1* mutations are reported; although very rare, these patients deserve a separated clinical management due to their peculiarities and response to specific (i.e., arsenic trioxide and ATRA) or conventional therapies.

In keeping with their established prognostic role across myeloid neoplasms, mutations in so-called “high molecular risk” (HMR) associated genes (*ASXL1*, *TP53*, *EZH2*, and *SRSF2*), have been shown to keep an adverse influence on survival in MPN-BP [6,27,28,29]. These studies are not always concordant in terms of representation of gene mutations at blast phase and even overtly conflicting for some aspects. Collectively, their findings are the basis for insight into leukemogenesis in MPN-BP and confirm the high level of complexity and genomic instability of this subset [28]. However, in current clinical practice, they do not drive major decisions. An intensive approach with a curative intent relies on mere progression from MPN, as crucial prognostic information, and patient eligibility for intensive treatment modalities.

## 4. Treatment

Currently there is no established standard of care for MPN-BP; treatment ranges from supportive care to low-intensity approaches, such as hypomethylating agents (HMA) or low dose cytarabine (LDAC), to more intensive strategies, that generally include induction chemotherapy or allogeneic hematopoietic stem cell transplant (HSCT). Treatment decision is mostly based on age, performance status, and competing comorbidities. On this basis, the adopted therapeutic strategies are discussed below by separating an intensive, curative approach from a low-intensity one.

### 4.1. Intensive Therapeutic Approach

#### 4.1.1. Chemotherapy

Table 2 summarizes the results of main retrospective studies available about results of standard intensive chemotherapy.

In a single-institution analysis at Mayo Clinic Rochester (Rochester, MN, USA) [16], approximately 40% of the 24 patients receiving intensive induction achieved a CR, and none showed resolution of bone marrow features of myelofibrosis.

A study by MD Anderson Cancer Center (MDACC) showed comparable results, i.e., overall response rate (including CR and CR with incomplete recovery (CRi)) was approximately 50%. This paper highlighted an important issue that only the patients undergoing consolidation with HSCT benefited in terms of survival after receiving induction [18].

In the Canadian report by Kennedy and colleagues, in the group of patients treated with curative intent, 18 (46%) achieved CR or CRi and 12 (31%) reverted to a chronic MPN phase. Survival was significantly improved for transplanted patients compared to those who responded to induction but were not transplanted (overall survival of 47% vs. 15% at 2 years, respectively, *p* = 0.03). The latter category had a prognosis superimposable to not-intensively treated patients [19].

As in the paper by MDACC, the meaningful prolongation of survival was achieved when HSCT was used as consolidation; chemotherapy alone did not provide relevant improvement in outcome over low-intensity therapy.

In a retrospective study by Mayo Clinic (Rochester, MN, USA), a total of 125 patients received chemotherapy that was considered to be more intensive than hydroxyurea; 69 were treated with AML-like induction chemotherapy, 26 with HMA, and 30 with other drugs, including investigational agents. The CR rate for AML-induction chemotherapy was 35%, with an additional 24% of patients achieving CRi (overall response rate of 59%). An interesting finding from this study was that the likelihood of obtaining CR/CRi was not undermined by adverse karyotype [24]. The results reiterated the role for HSCT as the option associated with the best outcome and suggested that the achievement of a response ameliorated even in not transplanted patients. The parallel analysis of a cohort from AGIMM (AIRC-Gruppo Italiano Malattie Mieloproliferative) led to analogue findings.

A recent Canadian study on 122 patients further emphasized the limited value of intensive chemotherapy in the absence of a HSCT program. When stratified according to general treatment strategy among supportive care, low dose or intensive therapy, no difference in outcome was seen for the latter two groups. A survival advantage became evident only by splitting the intensive treatment category according to HSCT. On the basis of their data, the authors concluded that patients without HSCT option could be spared from the toxicity related to aggressive treatment, in view of its virtually absent probability of a survival advantage [29].

CPX-351 is an encapsulation in nanoscale liposomes of cytarabine and daunorubicin at a synergistic 5:1 molar ratio [31]. This formulation has been tested initially in a phase 2, and then in a phase 3 randomized trial, compared to standard “3 + 7” in elderly (aged 60–75 y) patients with high-risk features, namely AML with myelodysplasia-related changes or therapy-related AML. CPX-351 produced a higher response rate (CR/CRi, 47.7% vs. 33.3%, *p* = 0.016), and longer OS (medians of 9.6 vs. 6 months, *p* = 0.005); however, results were similar after accounting for allogeneic HSCT. On the basis of these data, in 2017 FDA has approved CPX-351 for the treatment of the two above mentioned AML subsets. These results imply the potential use of CPX-351 also in the specific subset of MPN-BP, which was excluded from the clinical trials, and where the superiority of the drug compared to standard induction should be proved formally.

#### 4.1.2. Allogeneic Transplant

As previously stated, the application of HSCT is often the completion of a comprehensive treatment program; consistently, its results cannot be separated completely from previous therapies. Some relevant reports specifically address the issue of HSCT in MPN-BP and allow for some useful considerations.

A French study by Cahu et al. focused on the results provided by HSCT in a retrospective study gathering 60 cases, including 43 MPN-BP and 17 post MPN/MDS. As per disease status at HSCT, 37% of the cases were in CR1, while the majority (57%) had advanced disease (with seven cases transplanted upfront) [32]. It is also interesting to underline that a relevant fraction of patients (55%) underwent HSCT after reduced-intensity conditioning (RIC). Engraftment was achieved in 92% of patients and transplant-related mortality (TRM) was 22% at 3 years. Of note, CR status at HSCT was the only variable significantly associated with an improved DFS, and it maintained impact in multivariate analysis with conditioning type [32].

A smaller study by MDACC enucleated 14 patients receiving HSCT for leukemic transformation of myelofibrosis from a wider cohort of transplanted patients [22]. Although limited by sample size, this study confirmed the high rate of engraftment (100%) and further provided data about chimerism early after HSCT: almost 80% of patients had 100% donor myeloid and T cells at Day 30. More than half (seven out of 13) of the patients underwent HSCT not in CR; in spite of that, all patients achieved CR after transplant, in keeping with the high efficacy of the procedure, at least at early time-points.

The European Group for Blood and Bone Marrow Transplantation (EBMT) has retrospectively extrapolated 46 patients from the registry with post myelofibrosis AML undergoing HSCT. Ninety-one per-cent received induction as a bridge to HSCT (25% experienced a CR); 26 patients received a RIC regimen, while 20 a standard myelo-ablative one. In multivariate analysis, the only parameter associated with favorable outcome was CR status at the time of conditioning regimen starting [21]. The final outcome was significantly better for patients transplanted in CR; however, the authors emphasize that CR occurred in a minority (24%) of cases, while 50% turned out to be either refractory to induction or progressing at the time of HSCT. Interestingly, the benefit for CR patients was mainly due to lower TRM, whereas the incidence of relapse was identical.

In a retrospective study in the United States, the outcome of 19 allo-transplanted patients was reported to be affected by red blood cell transfusion dependence during the course of MPN, likely an indicator of worse overall clinical fitness [33]. Although not statistically significant, the authors underline a favorable impact of CR obtainment on outcome after HSCT.

In a retrospective analysis of the International Blood and Marrow Transplant Research (CIBMTR) database, the outcome of MPN-BP after HSCT appeared to be lower than that of de novo and post-MDS AML, and this difference emerged mainly in terms of relapse rate when the disease was transplanted into remission [34]. Once more, this finding highlights the clinical and biological uniqueness of MPN-BP.

#### 4.1.3. Summary of Available Evidence about Intensive Approaches

The first point is that intensive chemotherapy alone does not offer significant improvement in outcome over low-intensity therapy and it turns out to be useless if not followed closely by HSCT [18,19]. The latter is the only potentially curative approach, and it is able to provide long-term remission in 20–30% of patients, even when transplanted upfront or with refractory disease [21,22,30]. Therefore, when eligible for age and comorbidities, the crucial target of treatment should be the allocation to a HSCT program.

Regarding a bridge to HSCT, in spite of the absence of controlled clinical trials, and thus validated standards of care, conventional “3 + 7”/like induction chemotherapy (also including novel formulation as CPX-351) appears to be the best suitable option in MPN-BP subset. In de novo AML, this regimen has been adopted for some decades and it is still considered to be a standard of care. In fact, many efforts have been made to improve its results by increasing doses or adding drugs without clear advantages in overall outcome [35]. The categories of patients experiencing an improvement by intensification generally belong to favorable or intermediate risk groups, according to ELN stratification [26]. In adverse risk group, the increase in dose intensity is not able to overcome intrinsic resistance while it certainly accompanies a higher treatment-related toxicity. Having anticipated above the biological characteristics of MPN-BP, on the one hand, the transfer of data from de novo AML into this context reasonably leads to expect no advantages from the application of intensified chemotherapy schemes. On the other hand, a de-escalation of intensity, for instances based on HMA, has demonstrated to achieve scarce responses and its application is, therefore, better suitable for patients unfit for chemotherapy and HSCT, rather than eligible for a curative-intent approach. Novel drugs might be useful in the future as bridging strategies toward HSCT, by providing anti-leukemic efficacy with lower toxicity, but as per current knowledge, they cannot be recommended in a wider setting.

Regarding the role of disease status, many discussed reports reiterate how obtaining a response at the time of HSCT positively influences outcome. However, this occurs in a fraction of patients (collectively 20–30%) and the responses are virtually always transient. The data available from the literature show that patients transplanted upfront or with active disease can achieve durable remissions with HSCT (about 20% in reports by Japanese and EBMT groups [21,30]. As such, a therapeutic strategy that subordinates allogeneic HSCT to the response to induction chemotherapy appears inadequate because it excludes the vast majority of patients from the only curative option. That prompts one to rethink the timing of HSCT in this clinical context. A recent report by CIBMTR highlighted how the disease genotype can overcome the prognostic impact of disease status at transplant, in particular, a *TP53*-mutated status was correlated with dismal outcome regardless of the achievement of a response before HSCT [36]. Indeed, the immunologic effect by HSCT appears to be largely ineffective in this subgroup, possibly, also due to an immunosuppressive phenotype conferred by the *TP53* mutant [37]. Alternative therapeutic approaches should be explored for this category of patients, including the incorporation of a novel agent capable of restoring the normal TP53 conformation and function, such as APR-246 [38], and the introduction of maintenance strategies after HSCT in order to enhance the graft versus leukemia effect.

### 4.2. Low Intensity Therapeutic Approach

#### 4.2.1. Hypomethylating Agents

Hypomethylating agents (HMA), namely azacitidine and decitabine, are widely used in the setting of myelodysplastic syndromes and acute leukemias. Only a few reports have focused on MPN-BP and no controlled trials are available [17]. These studies are obviously downsized due to the limited number of patients enrolled, and often collect chronic and accelerated phases. Overall, the response rate is low, and the main advantage is based on a safer profile than standard intensive approaches. In a retrospective study by Mayo Clinic, 24 patients received HMA with a CR rate of less than 5% [24]. A recent trial by MDACC explored the combination of escalating doses of Ruxolitinib (up to 50 mg twice daily) with decitabine, concluding this approach to be feasible without any possible statements to be drawn about its efficacy, also due to the heterogeneity of inclusion criteria (including AP phases) [39]. In a retrospective study by Lancman et al. collecting several therapeutic strategies, four out of 27 (15%) receiving HMA obtained CR, some of them also allocated to HSCT after response [33]. In conclusion, HMA offer scarce chance of CR/CRi in sAML and, as such, do not appear suitable in a curative intent embedding HSCT. Rather they may be an option in unfit patients. In selected cases, preliminarily deemed as unfit and thus not eligible for transplant procedure, a fitness improvement deriving from a good response to treatment might lead to reconsider more intensive approaches later on.

#### 4.2.2. JAK1/2 Inhibitors

The rationale for the use of JAK2 inhibitor ruxolitinib in MPN-BP derives from the demonstrated efficacy at inducing reduction of spleen volume and amelioration of symptom burden in patients with MF and PV [39,40]. In MPN-BP subset the available experiences are scarce and generally limited by the fact that leukemic evolution often occurs under the selective pressure of JAK inhibition during the MPN phase.

Monotherapy with ruxolitinib at doses higher than those used for MF/PV (from 25 to 50 mg twice daily) was investigated in the context of MPN-BP (18 cases). A CR/CRi was obtained in three patients. According to preclinical data from murine models, the combination of ruxolitinib with HMA, namely decitabine, has been assessed in a U.S. phase 1 multicenter trial coordinated by MD Anderson Center (Houston, Texas, USA) [41]. The study showed that the combination therapy was feasible and suggested clinical efficacy, with an overall response rate of 53% and median overall survival of 7.9 months. In the phase 2 trial that followed, recruiting 25 patients, the main results were similar with an overall response rate and a median OS of 44% and 9.5 months, respectively (trial identifier NCT02076191) [42]. A French study has investigated the combination of ruxolitinib and intensive induction therapy (“3 + 7”) in a pilot study on six cases [43], obtaining two CR and two PR. The intent of the investigators relied on the idea that ruxolitinib might increase the depth of response, and thus serve as a better bridge to HSCT compared to chemotherapy alone.

#### 4.2.3. IDH Inhibitors

The advent of therapeutic agents able to target mutant IDH1/2 enzymes provided a new approach for patients bearing *IDH1/2* mutations, which occur at low frequency (2–4%) in the chronic phase but at higher incidence (up to 15%) at MPN-BP evolution, as reported in published retrospective series [44,45]. Enasidenib and ivosidenib are small molecule inhibitors of mutant IDH2 and IDH1, respectively; based upon results coming from phase 1/2 trials, they have been approved by the Food and Drug Administration (FDA) for relapsed or refractory AML associated with an *IDH* mutations, as well as first-line treatment for elderly (≥75 y) or unfit patients for ivosidenib. The European regulatory agency EMA has deemed the results of the uncontrolled studies carried out so far insufficient for their approval. The most common side effects included hyperbilirubinemia and a differentiation syndrome, with a later onset compared to the one observed in acute promyelocytic leukemia. These drugs are potential options for patients with secondary AML associated with *IDH1/2* mutations and are being tested within clinical trials, also in combinations with other agents (i.e., HMA and BCL-2 inhibitors). In a small retrospective series in MPN-BP, promising efficacy and safety was reported, obviously to be confirmed prospectively in larger cohorts [46].

#### 4.2.4. BCL-2 Inhibitors

The induction of apoptosis by BCL-2 inhibition is clearly emerging among the most promising therapeutic approaches in the field of AML. Venetoclax (VEN) is a potent, selective, oral inhibitor of BCL-2 which, in preclinical studies, demonstrated anti-leukemic activity as a monotherapy and additive properties in combination with HMA or LDAC. In AML setting, the drug has been tested in elderly patients with treatment-naive de novo or secondary AML ineligible for intensive chemotherapy and in relapsed/refractory (R/R) subset. Clinical data from phase 1/2 trials have led to FDA breakthrough designations for treatment combinations with HMA or LDAC in untreated patients with AML who are 75 years or older or who have comorbidities that preclude the use of intensive induction chemotherapy. The efficacy of the approach was confirmed in two prospective randomized multicenter trials in previously untreated patients, ineligible for standard induction therapy [47,48]. Responses were seen in all disease subsets with the highest benefit in patients bearing mutant *IDH1/2*, *NPM1*, or *FLT3* [47,48,49,50,51,52]. Patients with secondary AML, merging evolution from MDS, MPN, and MDS/MPN, had lower rates of response, although comparing well with historical data in this setting [53], thus, opening to a potential application of such treatment combinations in this disease category. Some preliminary retrospective series, specifically collecting patients with MPN-BP, reported encouraging results, although uncontrolled and still to be validated [54,55].

## 5. Conclusions

Evolution to blast phase is almost invariably a fatal event in the course of MPN disease. The long-lasting dynamics of the progression of the disease, passing through progressive and faster genetic steps, leads to the emergence of one or more neoplastic clones characterized by complex genetics, with all the clinical and prognostic correlates. Current therapeutic approaches are largely inadequate and unable to change the inherent adverse outcome. In this perspective, the application of targeted strategies should primarily aim to prevent the occurrence of a leukemic evolution. At the onset of MPN-BP, the development of effective combination treatments may represent a useful bridge to allogeneic transplant for eligible patients, as well as aid in prolonging the survival of unfit patients.

## Figures and Tables

**Table 1 jcm-10-00436-t001:** Response definitions according to post myeloproliferative neoplasms (MPN) acute myeloid leukemia (AML) consortium [20].

Category	Definition
Complete molecular response (CMR)	*Description*: Complete remission of both leukemia and MPN without detectable molecular markers associated with either leukemia or MPN *Hematologic profile*: ANC > 1000/uL; hemoglobin > 10 g/dL; Platelets > 100 × 10^9^/L; Absence of leukoerythroblastosis ^a^ *Spleen:* Non-palpable *Bone marrow*: Cellularity appropriate for age; resolution of abnormal morphology; blasts ≤ 5% ^b^; ≤ Grade 1 marrow fibrosis *Cytogenetics*: Normal karyotype ^c^ *Molecular markers*: Loss of any previously documented markers associated with either the leukemic or MPN clone ^d^
Complete cytogenetic response (CCR)	*Description*: Complete remission of both leukemia and MPN with detectable molecular markers associated with either leukemia or MPN *Hematologic profile*: ANC > 1000/uL; Hemoglobin > 10 g/dL; Platelets > 100×10^9^/L; Absence of leukoerythroblastosis ^a^ *Spleen*: Non-palpable *Bone marrow*: Cellularity appropriate for age; resolution of abnormal morphology; blasts ≤ 5% ^b^; ≤ Grade 1 marrow fibrosis *Cytogenetics*: Normal karyotype ^c^ *Molecular markers*: Residual expression of MPN/leukemia associated gene mutations (e.g., JAK2V617F, MPL515L/K) ^d^
Acute leukemia response-complete (ALR-C)	*Description*: Complete remission of leukemia with residual MPN features *Hematologic profile*: Absence of blasts ^a^ *Spleen*: < 25% increase in spleen size by palpation or imaging if baseline spleen < 10 cm or < 50% if baseline spleen ≥ 10 cm *Bone marrow*: Blasts ≤ 5% ^b^ *Cytogenetics*: Loss of cytogenetic abnormality associated with leukemic clone, may have persistent abnormality associated with MPN *Molecular markers*: Loss of any previously identified markers in leukemic clone, may have persistent molecular markers associated with MPN ^d^
Acute leukemia response-partial (ALR-P)	*Description*: Decrease in leukemic burden but without resolution of peripheral blood or bone marrow blasts and residual MPN features *Hematologic profile*: > 50% reduction in blasts *Spleen*: < 25% increase in spleen size by palpation or imaging if baseline spleen < 10 cm or < 50% if baseline spleen ≥ 10 cm *Bone marrow*: > 50% reduction in blasts *Cytogenetics*: No new abnormalities *Molecular markers*: No new abnormalities
Stable disease (SD)	*Description*: Failure to achieve at least ALR-P, but no evidence of progression for at least 8 weeks.
Progressive disease (PD)	*Description*: Progression of leukemia and/or background MPN *Hematologic profile*: For patients with 10–20% blasts: ≥ 50% increase to > 20% blasts; For patients with > 20% blasts: ≥ 50% increase to > 30% blasts; *Spleen*: > 25% increase in spleen size by palpation or imaging if baseline spleen < 10 cm and > 50% if baseline spleen ≥ 10 cm *Bone marrow*: For patients with 5–10% blasts: ≥ 50% increase to > 10% blasts; For patients with 10–20% blasts: ≥ 50% increase to > 20% blasts; For patients with > 20% blasts: ≥ 50% increase to > 30% blasts *Cytogenetics*: Does not apply *Molecular markers*: Does not apply

^(a)^ Absence of peripheral blood blasts by morphologic review of the peripheral smear on two occasions separated by at least 2 weeks. ^(b)^ Blast percentage can be assessed by morphologic review of aspirate and, in cases of punctio sicca, immunohistochemical staining of the marrow for CD34+, CD117+ is acceptable. ^(c)^ Normal karyotype by conventional cytogenetics in peripheral blood or bone marrow aspirate, if a cytogenetic abnormality is detected prior to treatment it must not be identified at time of assessment; if an abnormality is detected at baseline by Fluorescence in situ hybridization (FISH) it must be absent by FISH at time of assessment. ^(d)^ Absence or loss of evidence of mRNA transcript by quantitative PCR assay performed in a validated laboratory, this will also include any exploratory biomarkers determined to be positive prior to therapy. Emerging molecular methodologies (NGS, digital droplet PCR) are increasingly adopted in response monitoring in myeloid neoplasms.

**Table 2 jcm-10-00436-t002:** Summary of reports about intensive therapeutic approaches (including allogeneic hematopoietic stem cell transplant (HSCT)) in secondary acute myeloid leukemia (sAML).

Reference	Induction Chemotherapy				Allogeneic Transplant						
	*n*	Type	Response	OS, mo	*n*	Conditioning	Disease status	Donor	CIR @ 2y	NRM @ 2y	OS @ 2y
Mesa, 2005 [16]	24	“3 + 7” 75% HDAC 13% MEC 13%	CR 0%	3.9	-	-	-	-	-	-	-
Tam, 2008 [4]	41	Ida-HDAC 54% “3 + 7” 15%	CR/CRi 46%	NR	8	NR	CR 12.5% CRi 50% NR 37.5%	Sib 62.5% MUD 37.5	12.5%	12.5%	37.5%
Ciurea, 2010 [22]	-	-	-	-	14	MAC 36% RIC 64%	CR/CRi 43% NR 57%	Sib 57% MUD 43%	38%	29%	33%
Kennedy, 2013 [19]	38	“3 + 7” 66% MEC 32%	CR 32% CRi 5% c-MPN 26%	-	17	MAC 47% RIC 53%	CR/CRi 59% c-MPN 41%	Sib 70% MUD 30%	24%	47%	29%
Alchalby, 2014 [21]	-	-	-	-	38	MAC 53% RIC 47%	CR 23% NR 77%	Sib 45% MUD 55%	47%	28%	33%
Takagi, 2016 [30]	-	-	-	-	39	MAC 38% RIC 62%	CR 18% NR 52% Untreated 30%	Sib 21% MUD 38% CB 41%	34%	34%	29%
Tefferi, 2018 Mayo cohort [24]	66	“3 + 7”/like 90% Other 10%	CR 35% CRi 24%		24	NR	CR/CRi 67% NR 33%	NR	NR	NR	41%
Tefferi, 2018 AGIMM cohort [24]	48		CR 27% CRi 8%		25	MAC 76% RIC 24%	CR/CRi 40% NR 60%	Sib 40% MUD 44% Haplo 16%	39.5%	21.7%	41.5%

*n*, number of patients; HDAC, high dose cytarabine; MEC, mitoxantrone, etoposide, cytarabine; CIR, cumulative incidence of relapse; NRM, non-relapse mortality; OS, overall survival; CR, complete remission; CRi, complete remission with incomplete hematologic recovery; MAC, myeloablative condition; RIC, reduced intensity conditioning; Sib, sibling donor; MUD, matched unrelated donor; Haplo, haploidentical donor.

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
