# Peer review of "Acute Myeloid Leukemia Evolving from Myeloproliferative Neoplasms: Many Sides of a Challenging Disease"

_jcm, 2021, doi:10.3390/jcm10030436_

Round 1

Reviewer 1 Report

How to approach treatment of patient with MPN-BP is a major clinical challenge. The early detection of clonal alterations leading to AML development could enable to propose targeted strategies potentially improving outcome of patients. This review nicely illustrates these challenges.

Comments

1- Please provide a more exhaustive description of MPN molecular profile in the "2-biological feature" section. Beyond JAK2, CALR and MPL mutations, Grinfeld et al, NEJM 2018 proposed a molecular classifier predictive of clinical outcome and AML transformation. Spliceosome mutations are not discussed in this section.

2-The detection of mutations associated with AML transformation would require an exhaustive molecular analysis at diagnosis and during the follow up of patients with MPN. Today, NGS is not routinely performed for all MPN patients as JAK2, CALR and MPL hotspot mutations are commonly detected with allele specific methods.

According to this review, can the autor give some general recommendation for the use of high throughput sequencing at diagnosis and during the course of the treatment? Please discuss this point.

3-Table 1. The autors should precise the required sensitivity for MRD testing (d). Moreover, alternative strategies for "mRNA transcript by quantitative PCR" are available for molecular follow-up (ie NGS and digital PCR on genomic DNA)

Author Response

Reviewer 1

How to approach treatment of patient with MPN-BP is a major clinical challenge. The early detection of clonal alterations leading to AML development could enable to propose targeted strategies potentially improving outcome of patients. This review nicely illustrates these challenges.

Comments

1- Please provide a more exhaustive description of MPN molecular profile in the "2-biological feature" section. Beyond JAK2, CALR and MPL mutations, Grinfeld et al, NEJM 2018 proposed a molecular classifier predictive of clinical outcome and AML transformation. Spliceosome mutations are not discussed in this section.

I thank the reviewer for the comments, which in my opinion led to an improvement of the paper. I have updated the section 2 and the relative references according to the suggestion.

2-The detection of mutations associated with AML transformation would require an exhaustive molecular analysis at diagnosis and during the follow up of patients with MPN. Today, NGS is not routinely performed for all MPN patients as JAK2, CALR and MPL hotspot mutations are commonly detected with allele specific methods. According to this review, can the author give some general recommendation for the use of high throughput sequencing at diagnosis and during the course of the treatment? Please discuss this point.

Thanks for raising this point; I have added some general statements about the use of NGS in chronic and blast phase in Section 2.

3-Table 1. The authors should precise the required sensitivity for MRD testing (d). Moreover, alternative strategies for "mRNA transcript by quantitative PCR" are available for molecular follow-up (ie NGS and digital PCR on genomic DNA)

The Table 1 reports the formal recommendations by an international consortium for the assessment of response in MPN blast phase and as such it has been reported in the review. Specific requirements for sensitivity can’t be drawn in a definite way since a validated assay has not been developed for some molecular markers (for instance CALR mutants). According to the suggestion by the reviewer, we have added a comment about the implementation of modern techniques, such as ddPCR and NGS, in the legend of the Table.

Reviewer 2 Report

I want to congratulate the author on his excellent, comprehensive and well-written review. The tables are very helpful. The data are presented in an unbiased fashion and highlight the clinical challenges with MPN-BP patients. I only have the following minor comments:

1.) The author is eluding to the NCT02076191, which has been published recently https://pubmed.ncbi.nlm.nih.gov/33104796/)

2.) While I agree with the author that genetic factors are not ready to be used for clinical decision-making quite yet, there are definitely data supporting their use for prognostication following allo-HCT (e.g. https://pubmed.ncbi.nlm.nih.gov/33170935/). A discussion on the potential impact of more wide-spread use on practice patterns and outcomes as part of “future directions” section might be helpful. It seems like TP53 mutated patients do not benefit from HSCT but what else could be offered (APR-246?)

Author Response

I want to congratulate the author on his excellent, comprehensive and well-written review. The tables are very helpful. The data are presented in an unbiased fashion and highlight the clinical challenges with MPN-BP patients. I only have the following minor comments:

1.) The author is eluding to the NCT02076191, which has been published recently https://pubmed.ncbi.nlm.nih.gov/33104796/)

I really thank the reviewer for her/his comments about the paper.

As concerns the specific point, the above study was cited as ongoing but not quoted. We have now quoted it as reference number 43.

2.) While I agree with the author that genetic factors are not ready to be used for clinical decision-making quite yet, there are definitely data supporting their use for prognostication following allo-HCT (e.g. https://pubmed.ncbi.nlm.nih.gov/33170935/). A discussion on the potential impact of more wide-spread use on practice patterns and outcomes as part of “future directions” section might be helpful. It seems like TP53 mutated patients do not benefit from HSCT but what else could be offered (APR-246?)

I have modified the discussion on the role for disease status at allogeneic transplant, also quoting the paper by Gupta et al about the unfavorable impact of a TP53 mutated status. Accordingly, I have added a comment about the perspective of additional/alternative therapeutic strategies (including APR-246) for this disease subset.